# Study on the Treatment of ITP Mice with IVIG Sourced from Distinct Sex-Special Plasma (DSP-IVIG)

**DOI:** 10.3390/ijms242115993

**Published:** 2023-11-06

**Authors:** Wei Zhang, Xin Yuan, Zongkui Wang, Jixuan Xu, Shengliang Ye, Peng Jiang, Xi Du, Fengjuan Liu, Fangzhao Lin, Rong Zhang, Li Ma, Changqing Li

**Affiliations:** Institute of Blood Transfusion, Chinese Academy of Medical Sciences & Peking Union Medical College, Chengdu 610052, China; 18581556966@163.com (W.Z.); zongkui.wang@ibt.pumc.edu.cn (Z.W.); xjxventus@126.com (J.X.); shengliang.ye@ibt.pumc.edu.cn (S.Y.); peng.jiang@ibt.pumc.edu.cn (P.J.); xi.du@ibt.pumc.edu.cn (X.D.); fengjuan.liu@ibt.pumc.edu.cn (F.L.); fangzhao.lin@ibt.pumc.edu.cn (F.L.); rong.zhang@ibt.pumc.edu.cn (R.Z.)

**Keywords:** intravenous immunoglobulin, IVIG sourced from distinct sex-specific plasma (DSP-IVIG), immune thrombocytopenia, proteomics, macrophages

## Abstract

Intravenous immunoglobulin (IVIG) is a first-line drug prepared from human plasma for the treatment of autoimmune diseases (AIDs), especially immune thrombocytopenia (ITP). Significant differences exist in protein types and expression levels between male and female plasma, and the prevalence of autoimmune diseases varies between sexes. The present study seeks to explore potential variations in IVIG sourced from distinct sex-specific plasma (DSP-IVIG), including IVIG sourced from female plasma (F-IVIG), IVIG sourced from male plasma (M-IVIG), and IVIG sourced from a blend of male and female plasma (Mix-IVIG). To address this question, we used an ITP mouse model and a monocyte–macrophage inflammation model treated with DSP IVIG. The analysis of proteomics in mice suggested that the pathogenesis and treatment of ITP may involve FcγRs mediated phagocytosis, apoptosis, Th17, cytokines, chemokines, and more. Key indicators, including the mouse spleen index, CD16^+^ macrophages, M1, M2, IL-6, IL-27, and IL-13, all indicated that the efficacy in improving ITP was highest for M-IVIG. Subsequent cell experiments revealed that M-IVIG exhibited a more potent ability to inhibit monocyte phagocytosis. It induced more necrotic M2 cells and fewer viable M2, resulting in weaker M2 phagocytosis. M-IVIG also demonstrated superiority in the downregulation of surface makers CD36, CD68, and CD16 on M1 macrophages, a weaker capacity to activate complement, and a stronger binding ability to FcγRs on the THP-1 surface. In summary, DSP-IVIG effectively mitigated inflammation in ITP mice and monocytes and macrophages. However, M-IVIG exhibited advantages in improving the spleen index, regulating the number and typing of M1 and M2 macrophages, and inhibiting macrophage-mediated inflammation compared to F-IVIG and Mix-IVIG.

## 1. Introduction

Intravenous immunoglobulin (IVIG) is a mixture of immunoglobulins extracted from the pooled plasma of thousands of healthy donors. The utilization of plasma from multiple donors contributes to the broad spectrum and specificity of IVIG antibodies. IVIG is currently employed in more than 100 clinical indications, including the treatment of primary immunodeficiency disease (PID), secondary immunodeficiency syndrome (SID), autoimmune diseases (AIDs), etc. [1,2,3]. The main component of IVIG is immunoglobulin G (IgG). Following papain hydrolysis, classical IgG antibody consists of one molecule of fragment crystallizable (Fc) and two molecules of fragment of antigen binding (Fab) fragments [4,5].

AIDs arise from the body’s immune response against its own components, resulting in tissue damage and organ dysfunction. One such AID is ITP, characterized by platelet destruction mediated by autoantibodies [6]. The existing literature has identified numerous factors associated with the course of ITP, including macrophages, T helper cells (Th1, Th2, Th17) [7], cytotoxic T lymphocytes (CTL), regulatory T cells (Treg), T follicular helper cells (Tfh) [8,9,10], dendritic cells (DCs), megakaryocyte (MK), B cells, plasma cells, regulatory B cells (Breg) [11,12,13,14]. Additionally, internal homeostasis disorders of bone marrow (BM) have been implicated in ITP [8].

In clinical practice, IVIG serves as a first-line treatment drug for ITP [15,16,17,18,19], and the use of IVIG in the clinical treatment of severe ITP requires high-dose infusion [20,21]. IVIG production involves raw plasma, and it is used to treat various AIDs. But how does sex relate to these components? Silliman, C. C. and others found that female plasma expresses more pregnancy zone protein, coagulation factor V, complement factor H, C4B, etc., than male plasma, while male plasma expressed more Fc binding protein, transgelin-2, etc., than female plasma [22]. Moreover, the intervention of different sexual plasma in diseases also varied; for example, transfusion-related acute lung injury (TRALI), which is a high mortality adverse reaction caused by blood transfusion [23]. In 2006, American scholars suggested that pregnant women should be excluded from fresh frozen plasma (FFP) and platelet apheresis. The American Red Cross began to give priority to the distribution of plasma from male donors in 2007, and then the number of TRALI cases decreased by 80% [24]. Similarly, in the UK, starting from the end of 2003, 80% to 90% of FFP, platelets, and plasma in UK hospitals were sourced from male donors. It was found that after 2003 in the UK, the reports of TRALI showed a decreasing trend year by year [25].

In addition, Desai et al. found that the prevalence of AIDs varied between males and females [26]. For example, women were more likely to experience multiple systemic lupus erythematosus (SLE), rheumatoid arthritis (RA), multiple sclerosis (MS), and Sjögren’s syndrome (SS) than men. These differences may arise from a multitude of factors, including susceptibility related to sexual target organs, reproductive function, hormone levels, environmental factors, immune responses, and genetic factors [27].

IVIG undergoes a series of standardized tests after preparation, including ensuring an IgG content of over 95%, distribution of IgG subclasses, Fc function, prekallikrein activator (PKA) level, hemagglutinin, and more [28]. Despite these quality controls, there remain significant variations in the antibody titers against crucial pathogens in IVIG products from different companies [29,30,31]. Additionally, even within the same company, different batches of IVIG can exhibit varying antibody titers [32,33]. M Sphere et al. found that IVIG products from different manufacturers had different complement clearance abilities [34]. Furthermore, the level of sialic acid in the Fc fragment of IVIG varies significantly among different companies [3]. Beyond these factors, variations in the source of raw plasma, influenced by endemic diseases and vaccination status, contributed to the variability of IVIG [28,35]. Thus, to ensure the preparation of high-quality IVIG products and maintain consistency, a systematical optimization of quality design, including the quality control of raw plasma, is imperative.

China’s plasmapheresis stations are primarily located in rural areas where the female population significantly outnumbers the male population, with female plasma donors outnumbering male donors by a factor of 2.4 [36]. Consequently, the majority of China’s IVIG raw plasma is sourced from female donors.

In the context of ITP, platelets are conditioned by autoantibodies and are primarily cleared by splenic macrophages. Macrophages express CD14 and CD36 and can be classified into classically activated macrophages (M1) and alternatively activated macrophages (M2). M1 macrophages are mainly activated by TNF-α, IFN-γ, and lipopolysaccharide (LPS), contributing to pro-inflammatory, anti-tumor, and anti-microbial functions. M1 macrophages highly express CD68 and CD86 and can release high levels of pro-inflammatory cytokines, such as TNF-α, IL-6, IL-1β, and nitric oxide synthase (iNOS) [37]. On the other hand, M2 macrophages are mainly produced by the activation of IL-4, IL-13, and IL-10. M2 macrophages release anti-inflammatory cytokines, such as IL-10 and TGF-β, which exerts anti-inflammatory and immunosuppressive effects. M2 cells are characterized by the expression of CD163 and CD206 [37]. An imbalance in the M1/M2 ratio has been reported in ITP patients, contributing to an increase in pro-inflammatory cytokines and inadequate immune response [1,7,37,38,39]. Following IVIG treatment, macrophage number and M1 subtype decrease while M2 subtype increases, but M2 growth is not always observed [40]. Therefore, M1 and M2 macrophages play crucial roles in ITP, influencing its pathogenesis, inflammation, and immune response.

The interaction between the Fc domain of IgG in IVIG and Fcγ receptors (FcγRs) triggers effector cells to exert immune regulatory effects. In humans, FcγRs include FcγRI (CD64), FcγRII (CD32), FcγRIII (CD16). FcγRII can be further divided into FcγRIIa, FcγRIIb, and FcγRIIc [41]. According to function, human FcγRs can be divided into active FcγR with immunoreceptor tyrosine activating motif (ITAM) and inhibitory FcγR with immunoreceptor tyrosine inhibitory motif (ITIM) [41]. FcγRIIb is the only inhibitory Fc in humans. The expression of FcγRs is regulated by cytokines. Pro-inflammatory cytokines typically increase the expression of activation of active FcγRs, while anti-inflammatory signals can downregulate active FcγRs and enhance FcγRIIb expression [42,43,44]. Unlike humans, mouse FcγRs include FcγRI (CD64), FcγRIIb (CD32b), FcγRIII (CD16), and FcγR IV. Significantly, mice only have FcγRIIb, with no FcγRIIa and FcγRIIc.

In conclusion, sexual differences exist in plasma proteins, potentially affecting the probability of transfusion-related acute lung injury (TRALI) between males and females. Additionally, sexual differences also exist in the incidence and outcome of AIDs. Given the prevalence of female plasma donors in China and the variability in clinical IVIG efficacy, the question arises: can sexual differences in raw plasma impact the efficacy of IVIG in treating AIDs? To address this question, we established an ITP mouse model treated with DSP IVIG, conducting comprehensive analyses encompassing blood routine examination, proteomics analysis, assessments of mouse immune cells (M1/M2 macrophages and its FcγRs expression, Th, CTL, Th1, Th2, Th17), and cytokine secretion. Meanwhile, we utilized a monocyte–macrophage inflammation model to investigate the ability of DSP-IVIG to inhibit phagocytosis. Specifically, THP-1 cells were differentiated into M1 and M2 macrophages (THP-1-M1/M2) with or without IVIG. The effects of DSP-IVIG on M1/M2 macrophage cell apoptosis, cell typing, and FcγRs were then investigated. Finally, we tested the ability of DSP-IVIG to activate complement and the binding ability to FcγRs on the surface of THP-1.

## 2. Results

### 2.1. Mouse Blood Routine Test

The mouse model of ITP was established according to the methodology outlined in Figure 1A. This study involved six groups of mice. In detail, PBS represents the mice injected with PBS and albumin. AB was the mice given purified rat anti-mouse CD41 (Clone: MWReg30) and albumin. WG, F-IVIG, M-IVIG, and Mix-IVIG represent the mice injected with MWReg30 and WG IVIG (IVIG from Shenzhen Weiguang Biological Products Co., LTD, Shenzhen, China), F-IVIG, M-IVIG, and Mix-IVIG, respectively. When the mice were euthanized on D7 (day 7), it was found that the spleen tissue of the AB group displayed enlargement compared to the PBS group, whereas IVIG treatment resulted in the recovery of the spleen tissue (Figure 1B). The AB group displayed significant decreases in PLT (platelet), RBC (Red Blood Cell), HGB (Hemoglobin), HCT (Hematocrit), and PCT (Plateletcrit) (*p* < 0.0001); additionally, it showed increases in spleen index, RDW-SD (Red Blood cell Distribution Width-SD), MPV (Mean Platelet Volume), PDW (Platelet Distribution Width), and P-LCR (Platelet-Large Cell Ratio) (*p* < 0.0001). Notably, these alterations significantly reversed following IVIG treatment (*p* < 0.0001) (Figure 1C–O).

### 2.2. Proteomics Analysis of Mouse Spleen Tissue and Plasma

To analyze the differentially expressed proteins (DEPs) among the six groups of mice, we conducted proteomics studies on both spleen and plasma samples. DEPs were identified using the criteria of fold change (FC) >1.5 times (upregulated) or <0.67 times (down-regulated) with a *p*-value < 0.05. The results were visually represented using column graphs (Figure 2A,B).

Compared to the PBS group, the AB group exhibited significant enrichment in biological process (BP) categories such as negative regulation of endothelial and epithelial cell apoptotic processes, negative regulation of extrinsic apoptotic signaling via death domain receptors, blood coagulation, and fibrin clot formation. In comparison to the AB group, the M-IVIG group displayed enrichment in BP terms related to immune responses, immune system processes, interferon-gamma-mediated signaling pathways, responses to IFN-γ and IFN-β, cell surface receptor signaling pathways, positive regulation of cytokine-mediated signaling, regulation of immune responses, etc. Compared to the Mix-IVIG group, BP analysis of the M-IVIG group indicated associations with macrophage homeostasis, response to IL-13, regulation of definitive erythrocyte differentiation, etc. (Figure 2C–E).

KEGG analysis was also performed on the DEPs. Compared to the PBS group, the upregulated proteins in the AB group were involved in the toll-like receptor signaling pathway, NF-kappa B signaling pathway, autophagy—animal, apoptosis, endocytosis, lysosome, etc., whereas the downregulated proteins were involved in chemokine signaling pathway, cytokine–cytokine receptor interaction, complement and coagulation cascades, etc. Upregulated proteins in the F-IVIG group compared to the AB group showed that platelet activation, chemokine signaling pathway, PI3K-Akt signaling pathway, and apoptosis were enriched, while downregulated proteins were related with antigen processing and presentation, endocytosis, NF-kappa B signaling pathway, toll-like receptor signaling pathway, etc. Compared to the Mix-IVIG group, the upregulated proteins in the F-IVIG group were enriched in the chemokine signaling pathway, cytokine–cytokine receptor interaction, and platelet activation; moreover, the downregulated proteins were associated with Th17 cell differentiation, progesterone-mediated oocyte maturation, necroptosis, antigen processing and presentation, estrogen signaling pathway, etc. Compared to the Mix-IVIG group, downregulated proteins in the M-IVIG group were mainly involved in necroptosis, Th17 cell differentiation, antigen processing and presentation, and estrogen signaling pathway. Compared to the M-IVIG group, upregulated proteins in the F-IVIG group were applied to apoptosis, leukocyte transendothelial migration, phagosome, etc. (Figure 2F–J).

### 2.3. ITP Mouse Spleen Immune Cells

Compared to the PBS and AB groups, the IVIG treatment groups showed a significant decrease in CD64^+^ macrophages (*p* < 0.01). Particularly, the number of CD64^+^ cells in the F-IVIG group was obviously more than that in the M-IVIG and Mix-IVIG groups (*p* < 0.01) (Figure 3A). Meanwhile, CD32b^+^ cells increased in the IVIG-treated groups (*p* < 0.01), and CD32b^+^ cells in the Mix-IVIG group significantly increased compared with F-IVIG and M-IVIG groups (*p* < 0.0001) (Figure 3B). Moreover, Mix-IVIG treatment resulted in more CD16^+^ cells than M-IVIG treatment (*p* < 0.05) (Figure 3C).

Subsequently, we investigated the effects of IVIG on M1/M2 macrophages. Compared to the PBS group, Mix-IVIG treatment significantly increased monocytes when compared with F-IVIG and M-IVIG groups (*p* < 0.05) (Figure 3D). CD11b^+^ monocytes in M-IVIG and Mix-IVIG groups were elevated compared with those in the F-IVIG group (*p* < 0.01) (Figure 3E). For M1, Mix-IVIG was greater than F-IVIG, which was greater than M-IVIG (ns) (Figure 3F). For M2, M-IVIG was greater than F-IVIG, while F-IVIG was greater than Mix-IVIG (ns) (Figure 3G).

IVIG treatment showed no effects on the counts of splenocyte Th1 (IFN-γ+) cells, Th2 (IL-4+) cells, and Th17 cells (IL-17A+) (Figure 3H,I,K). However, IVIG significantly downregulated the proportion of Th1/Th2 (Figure 3I).

### 2.4. ITP Mouse Cytokines

Compared to the PBS group, IL-13, IL-10, IL-1β, and MCP-1 of the AB group were downregulated (*p* < 0.05), whereas IFN-γ was upregulated (*p* < 0.05). IVIG treatment resulted in the upregulation of IFN-γ (*p* < 0.0001), IL-1α (*p* < 0.01), IL-17 (*p* < 0.05), and IP-10 (*p* < 0.05) (Figure 4). Among the DSP-IVIG groups, the cytokines IL-13, IL-27, IL-6, and IL-12P70 exhibited differing expression patterns, with the M-IVIG group > Mix-IVIG group (*p* < 0.05).

### 2.5. Inhibition of THP-1 and M1/M2 Phagocytosis by DSP-IVIG

After lysing unphagocytized sensitized erythrocytes, we observed the phagocytosis of sensitized erythrocytes by THP-1 cells under the microscope (Figure 5A). At IVIG concentrations of 1 μg/mL, 0.5 μg/mL, and 0.2 μg/mL, M-IVIG exhibited a stronger ability to inhibit phagocytosis than Mix-IVIG (*p* < 0.05) (Figure 5B).

We induced the differentiation of THP-1 into M1 and M2 macrophages by adding different cytokines (Figure 5C). We found that the addition of both Fc and IVIG resulted in a decrease in CFSE+ macrophages (*p* < 0.0001), suggesting that both Fc and IVIG can inhibit macrophage phagocytosis of sensitized erythrocytes. For THP-1_M1, the CFSE+ cell in F-IVIG was greater than M-IVIG (*p* < 0.05), while Mix-IVIG group was bigger than M-IVIG (*p* < 0.01). Consequently, M-IVIG exhibited a more significant inhibitory effect compared to F-IVIG (*p* < 0.05) and Mix-IVIG (*p* < 0.01) (Figure 5D).

### 2.6. Cell Apoptosis

For THP-1_M2, necrotic cells in M-IVIG and Mix-IVIG were greater than F-IVIG (*p* < 0.01) (Figure 5E). Meanwhile, the number of live cells in F-IVIG was greater than those of M-IVIG and Mix-IVIG (*p* < 0.01) (Figure 5G).

### 2.7. M1/M2 Macrophage Markers

The expression levels of CD36 and CD68 on THP-1_M1 cells treated with F-IVIG were significantly higher than those on cells treated with M-IVIG (*p* < 0.05) (Figure 6A,B). Notably, the presence of sensitized erythrocytes led to an upregulation of CD206 and CD86 in macrophages, while IVIG treatment resulted in a downregulation of CD206 and CD86 (Figure 6C,D).

### 2.8. FcγRs Expression

For THP-1_M1/M2, live cells of Mix-IVIG were higher than that of F-IVIG and M-IVIG (*p* < 0.05) (Figure 6E). For THP-1_M1, CD64^+^ cells of F-IVIG and M-IVIG were higher than Mix-IVIG (*p* < 0.01). For THP-1_M2, CD64^+^ cells of F-IVIG were bigger than that of M-IVIG and Mix-IVIG (*p* < 0.0001) (Figure 6F). For THP-1_M1, after adding sensitized erythrocytes, CD32^+^ cells were downregulated, and IVIG can improve this downregulation. At the same time, CD32^+^ cells of F-IVIG were more than those of Mix-IVIG and M-IVIG (*p* < 0.05) (Figure 6G). For THP-1_M1/M2, after adding sensitized erythrocytes, CD16^+^ cells upregulated, and IVIG can improve this upregulation. For THP-1_M1, CD16^+^ cells of F-IVIG were more obvious than that of M-IVIG (*p* < 0.05) (Figure 6H).

### 2.9. Complement Activation by the Fc Segment of DSP-IVIG

The ability of IVIG to activate complement showed that Mix-IVIG exhibited a higher capacity compared to M-IVIG (*p* < 0.05) (Figure 7A).

### 2.10. Binding Capacity of Fc Segment of DSP-IVIG for FcγRs

The capacity of the Fc segment of DSP-IVIG to bind FcγRs revealed that M-IVIG showed a stronger binding capacity than F-IVIG (*p* < 0.05) (Figure 7B).

## 3. Discussion

In our study, we found that M-IVIG can better improve the spleen index of ITP mice. M-IVIG can better regulate the number and typing of M1 and M2 macrophages in order to exert the anti-inflammatory effect of IVIG. Specifically, the expression of monocytes, M1, M2, and FcγRIII (CD16) and the secretion of IL-6, IL-27, and IL-13 suggested that M-IVIG was better than F-IVIG and Mix-IVIG in improving ITP. Meanwhile, in cell experiments, we found that within a certain concentration range of IVIG, the phagocytosis of THP-1 on sensitized erythrocytes of M-IVIG was weaker; that is, M-IVIG may show weaker phagocytosis mediating by autoantibodies. For the cellular apoptosis experiment, there were more necrotic M2 cells and fewer M2 living cells in the M-IVIG group, which made M2 with weaker phagocytosis. M-IVIG can better downregulate CD36, CD68, and CD16 of M1, indicating that M-IVIG may exert a better anti-inflammatory effect by inhibiting the pro-inflammatory function of M1. M-IVIG had a stronger ability to bind FcγRs of THP-1 cells. The ability to activate the complement of M-IVIG was weaker, which can reduce the inflammatory response caused by excessive activation of the complement.

First, the proteomics data obtained from spleen and plasma in mice suggested that IVIG treatment for ITP should primarily focus on modulating the differentiation of T cell lines, T-cell receptor binding, IL-17 signaling pathway, FcγRs mediated phagocytosis, cytokines, chemokines, etc. These findings were consistent with the work of Wang et al. [45], Ding et al. [46], and Segú-Vergés et al. [47]. In addition, Wang et al. reported that ITP samples exhibited higher counts of macrophages and Th cells compared to normal samples, while the counts of activated CD8+ T cells, mast cells, and plasmacytoid dendritic cells were lower [45]. These findings were in line with our proteomics data from ITP mice.

Then, the analysis of blood routines in mice revealed that all three types of DSP-IVIG effectively improved ITP. Specifically, mice treated with Mix-IVIG exhibited a larger spleen compared to those treated with M-IVIG, suggesting that M-IVIG was more effective in improving the spleen index of ITP mice. Additionally, the expression levels of FcγRIII (CD16), IL-6, IL-27, and IL-13, and the numbers of monocytes and M1 and M2 macrophages showed a consistent pattern of improvement with DSP-IVIG, where M-IVIG outperformed Mix-IVIG (*p* < 0.05). However, the data pertaining to CD11b, FcγRI (CD64), FcγRIIb (CD32b), and CTL count were inconsistent. These findings suggested that the factors influencing the efficacy of IVIG in the treatment of ITP were multifaceted, which may explain the observed differences in the recovery of spleen index. The results of M1 and M2 macrophages in mouse spleen tissue showed that, in ITP mice, monocytes and M1 macrophages increased while M2 macrophages decreased. Moreover, these changes were partially reversible upon administration of IVIG. These findings were consistent with previous studies. For example, Di Paola et al. found that macrophages obtained from ITP patients expressed more M1-specific iNOS and fewer M2-specific CD206 compared to healthy individuals. Additionally, macrophages obtained from ITP patients secreted more IL-6, TNF-α, and IFN-γ and less IL-4 and IL-10 compared to healthy individuals [37]. M1 and M2 macrophages play crucial roles in the pathogenesis of ITP and contribute to impaired inflammation and immune response [7,48,49]. A comparative study involved in the phenotype and function of splenic macrophages between 24 control individuals and 86 ITP patients found that the expression of FcγRs between ITP and control individuals showed no difference. However, the phagocytic function of splenic macrophages decreased in ITP patients receiving IVIG treatment within 3 weeks before splenectomy [50]. Okamoto et al. found that after immunohistochemical staining of bone marrow (BM) tissue in ITP patients, there were more cells expressing CD68, CD163, and IL-17 in the bone marrow of ITP patients than in the control group [51]. Wu et al. reported a significant reduction in the expression of FcγRIIB (CD32b) on spleen macrophages in ITP patients [42]. All of these findings were consistent with our results.

In the present study, it was observed that M-IVIG exhibited advantages in regulating the spleen index, as well as the number and type of M1 and M2 macrophages in ITP mice. Subsequently, THP-1 cells were induced to differentiate into M1 and M2 macrophages, and it was found that the groups with sensitized erythrocytes exhibited significant upregulation of CD36 and CD68 in M1 macrophages compared with the group with unsensitized erythrocytes. However, both CD36 and CD68 were downregulated after the administration of IVIG. This suggested that IVIG can inhibit the inflammatory response by downregulating the proportion of M1 macrophages. These results were consistent with the findings in our mouse experiment that IVIG improved ITP by downregulating M1 macrophages.

Human FcγRs play a crucial role in immune function. The balance between activated FcγRs and FcγRIIb is vital in disease status, reflecting a balance or imbalance in the opposite functions of these receptors. FcγRs-mediated signal transduction significantly impacts the function of monocytes and macrophages, which is a key determinant of macrophage polarization [52,53]. Norris et al. found that anti-GPIIb/IIIa autoantibody-mediated FcγRs-dependent phagocytosis of splenic macrophages depended on FcγRI and FcγRIII. However, there were no significant differences between the ITP patients and the control group in the spleen B cells (CD19), T cells (CD3), monocyte/macrophages (CD14), and granulocytes (CD66b). Meanwhile, the expression of macrophages FcγRs (FcγRI, FcγRIIa, FcγRII, FcγRIII) increased (no significant statistical difference) [54]. These findings were partly consistent with our study; thus, after adding sensitive erythrocytes, the changes in CD64 of THP-1_M1 and THP-1_M2 were not significant. However, after IVIG administration, CD64 was unregulated. This suggested that IVIG may not exert immunosuppressive effects by downregulating CD64. However, IVIG can downregulate CD16 and CD32 of THP-1_M1, indicating that IVIG might regulate macrophage function by altering FcγRs of M1 and M2 macrophages.

Concerning complement activation, Cristina Segú-Vergés et al. found that diseases with good therapeutic effect of IVIG were characterized by an imbalance of B cells and complements system, suggesting a close relationship between IVIG and complement system [47]. Increasing evidence indicates that complement inhibition is a key anti-inflammatory mechanism of IVIG. For example, ecuzumab, a monoclonal antibody that inhibits the activation of terminal complement by binding to C5, has been approved for the treatment of Myasthenia gravis (MG) [55] and multifocal motor nerve neuropathy (MMN) [56]. Additionally, C3 and C4B are related to the mechanisms of action of IVIG, directly neutralizing these complement proteins [57,58]. In addition, IVIG can block C3 activation, bind to C1q, and prevent pathogenic antibodies from triggering complement cascade reactions [58,59,60]. IgG can neutralize complement-activating proteins (including C3a and C5a), and high levels of IVIG can inhibit C3b and C4b uptake to the cell surface, thereby preventing complement-mediated tissue damage [59,61]. These mechanisms explain how IVIG can prevent complement-mediated tissue damage [62]. In addition, M Sphere et al. found in vitro experiments demonstrating that different IVIG products had varying abilities to activate complement [34]. In our study, M-IVIG showed a weaker ability to activate complement, indicating a stronger ability to prevent complement-mediated damage. This difference may be one reason why M-IVIG can better improve the spleen index of ITP mice.

Naturally, there were several limitations in this study. Firstly, it is worth noting that the mouse ITP model may not entirely mimic the intricate immune milieu observed in ITP patients due to its inadequate inflammatory state [63]. Furthermore, the exclusive emphasis on female mice introduces ambiguity regarding the immune regulatory effect of DSP-IVIG on male ITP mice.

## 4. Materials and Methods

### 4.1. Mouse Blood Routines

SPF grade Balb/c female mice, 6–8 weeks old, weighing 20 g, were selected from Chengdu dossy experimental animals co. LTD of China. BD Pharmingen™ Purified Rat Anti-Mouse CD41 (Clone: MWReg30, BD Pharmingen, Franklin Lakes, NJ, USA) was used to induce ITP in mice according to a dose-escalation approach [64,65,66,67]. In the present study, a commercially available IVIG (5%, 50 mL, batches of 20200619J) obtained from Shenzhen Weiguang Biological Products Co., LTD (namely WG) was used as a positive control. F-IVIG, M-IVIG, and Mix-IVIG (5%, 50 mL, batches of 20210511, 20210512, and 20210513, respectively) were prepared by Shenzhen Weiguang Biological Products Co., LTD according to the standard protocols. To mimic the high-dose IVIG treatment used in severe ITP cases, we administered IVIG intervention from Day 2 (D2) to D7 via intraperitoneal injection of 800 μL of IVIG with a concentration of 50 mg/mL, adjusted according to the weight of the mice. IVIG was consistently administered 2 h after each MWReg30 injection. We established six groups of mice for this experiment (see Table 1 for details).

On D7, the mice were weighed. The mice were then anesthetized using 2,2,2-Tribromoethanol (Sigma-Alderich, St. Louis, MO, USA). Blood was collected via the inferior vena cava for blood routine testing using an automated animal blood analyzer (SYSMEX XT-1800i, Sysmex Corporation, Kobe, Japan). After centrifugation at 4 °C for 15 min at 3000× *g*, the upper plasma of the blood samples was collected for subsequent testing. The spleens were carefully removed and weighed. The mouse spleen index was calculated (spleen weight (mg)/body weight (g) × 10). The spleens were then divided and stored at −80 °C for further testing.

### 4.2. Proteomics of Mouse Spleen Tissue and Plasma

Data Independent Acquisition (DIA) technology was used to conduct proteomics identification of mouse spleen tissue and plasma. In brief, it included protein extraction and peptide enzymolysis, filter-aided sample preparation (FASP) digestion, DIA mass spectrometry separation, GO enrichment analysis, and KEGG pathway annotation [68].

### 4.3. Detection of Splenic Immune Cells in Mice by Flow Cytometry

The spleen was ground until the red block structure was no longer present, and splenocytes were collected after removing erythrocytes using lysing buffer (BD Biosciences, Franklin Lakes, NJ, USA), followed by resuspending by PBS with 2% FBS. Splenic cells (1 × 10^6^ of cells) were used for subsequent flow cytometer detection. Some markers (IFN-γ, IL-4, and IL-17a) required stimulation culture before detection; that is, 2 μL of leukocyte activation cocktail (BD Pharmingen, Franklin Lakes, NJ, USA) was added to each 1 × 10^6^ of splenocytes. Subsequently, the cells were cultured at 37 °C in a 5% CO_2_ cell incubator for 6 h. Then, we collected cells and appended 1 μL of fixable viability stain780 (BD Horizons, Franklin Lakes, NJ, USA) to detect cell viability. Sometimes, non-specific binding needed to be blocked; that is, cells were washed and resuspended using PBS with 1% BSA. If necessary, 1.6 μL of purified rat anti-mouse CD16/CD32 (Mouse BD Fc Block^TM^, BD Pharmingen, Franklin Lakes, NJ, USA) was used to block. Then, surface staining antibodies were added and stained at 4 °C for 30 min in the dark. After washing the cells, cell fixation was performed if necessary, followed by intracellular or nuclear marker staining and flow cytometer detection (BD FACSCelesta, BD Biosciences, Franklin Lakes, NJ, USA). Antibodies were used as follows: To detect FcγRs, 2 μL of FITC Rat Anti-Mouse CD11b (BD Pharmingen, Franklin Lakes, NJ, USA), 5 μL of BV605 Rat Anti-Mouse F4/80 (BD OptiBuild, Franklin Lakes, NJ, USA), 5 μL of APC Anti-Mouse CD32b (eBioscience, San Diego, CA, USA), 5 μL of PE CD16 recombinant rabbit monoclonal antibody (Invitrogen, Waltham, MA, USA), and 5 μL of Percp-eFluor710 Anti-Mo CD64 (eBioscience, San Diego, CA, USA) were used. For M1/M2 detection, 2 μL of FITC Rat Anti-Mouse CD11b (BD Pharmingen, Franklin Lakes, NJ, USA), 5 μL of BV605 Rat Anti-Mouse F4/80 (BD OptiBuild, Franklin Lakes, NJ, USA), 5 μL of BV421 Rat Anti-Mouse CD86 (BD Horizon, Franklin Lakes, NJ, USA), and 5 μL of Alexa Fluor647 Rat Anti-Mouse CD206 (BD Pharmingen, Franklin Lakes, NJ, USA) were added. With regard to CTL/Th, 5 μL of PercP-Cy5.5 Hamster Anti-Mouse CD3 (BD Pharmingen, Franklin Lakes, NJ, USA), 2 μL of FITC Rat Anti-Mouse CD4 (BD Pharmingen, Franklin Lakes, NJ, USA), and 5 μL of APC Rat Anti-Mouse CD8a (BD Pharmingen, Franklin Lakes, NJ, USA) were appended. For Th1/2/17 tests, 5 μL of PercP-Cy5.5 Hamster Anti-Mouse CD3 (BD Pharmingen, Franklin Lakes, NJ, USA), 2 μL of FITC Rat Anti-Mouse CD4 (BD Pharmingen, Franklin Lakes, NJ, USA), 5 μL of APC Rat Anti-Mouse IFN-γ (BD Pharmingen, Franklin Lakes, NJ, USA), 5 μL of PE-Cy7 Rat Anti-Mouse IL-4 (BD Pharmingen, Franklin Lakes, NJ, USA), and 5 μL of PE Rat Anti-Mouse IL-17A (BD Pharmingen, Franklin Lakes, NJ, USA) were added. Transcription Factor Buffer Set (BD Biosciences, Franklin Lakes, NJ, USA), BD Cytofix/Cytoperm^TM^ Fixation/Permeabilization Kit (BD Biosciences, Franklin Lakes, NJ, USA), and eBioscience^TM^ FOXP3 Transcription Factor Staining Buffer Set (Invitrogen, Waltham, MA, USA) were also used. The indicators with low expression levels were stained with compensation microspheres, BD™ CompBeads Anti-Rat Ig, κ/Negative Control Compensation Particles Set (BD Biosciences, Franklin Lakes, NJ, USA) and BD™ CompBeads Anti-Mouse Ig, κ/Negative Control Compensation Particles Set (BD Biosciences, Franklin Lakes, NJ, USA), to distinguish the positive and negative groups.

### 4.4. Detection of Mouse Cytokines

The LXSAMSM-17 Mouse Premixed Multi-Analyte Kit (R&D, Minneapolis, MN, USA) was used to detect 17 factors (MCP-1, IP-10, IFN-γ, IL-1β, IL-4, IL-6, IL-12 P70, IL-17, TNF-α, MIP-1β, GM-CSF, IL-1α, IL-2, IL-5, IL-10, IL-13, and IL-27) in mouse splenic samples following the instructions.

### 4.5. Inhibition of THP-1 and M1/M2 Phagocytosis by DSP-IVIG

Human acute monocyte leukemia cell line THP-1 (TIB-202, Lot:70029994, ATCC, Manassas, VA, USA) was cultured using RPMI medium (Gibco, Billings, MT, USA) supplemented with 2-mercaptoethanol (Gibco, Billings, MT, USA) to a final concentration of 0.05 mM, 10% (*v*/*v*) fetal bovine serum (ExCell Bio, Shanghai, China), and 1% (*v*/*v*) of penicillin–streptomycin (HyClone, Logan, UT, USA). Subsequently, the cell suspension was supplemented with 25 ng/mL of PMA (Sigma, St. Louis, MO, USA) to make the final concentration of 1 × 10^6^ cells/mL and then gently mixed and cultivated for 24 h. Afterward, we needed to replace the supernatant with fresh medium to cultivate for another 24 h. To induce the differentiation of M0 macrophages to M1 macrophages, 10 pg/mL of LPS (Sigma, St. Louis, MO, USA), 20 ng/mL of IFN-γ (PeproTech Inc., Cranbury, NJ, USA), and 20 ng/mL of IL-6 (PeproTech Inc., Cranbury, NJ, USA) were added. Similarly, to induce M0 to M2 type macrophages, 20 ng/mL of IL-4 (200-04, PeproTech Inc., Cranbury, NJ, USA), IL-13 (PeproTech Inc., Cranbury, NJ, USA), and IL-6 were added. The cells were then cultured in a 37 °C, 5% CO_2_ incubator for 48 h. After 24 h of differentiation stimulation, either 50 μg/mL of IVIG or 10 μg/mL of Fc (10702-HNAC, Sino Biological Inc., Houston, TX, USA) was added.

An amount of 3 mL of RhD-positive fresh peripheral blood anticoagulated with sodium citrate was collected following centrifuging at 2000× *g* for 5 min. An amount of 50 μL of hematocrit erythrocytes was aspirated and washed with pre-cooled PBS 5 times following the addition of 5 μL of 10 mm CFSE (BD pharmingen, Franklin Lakes, NJ, USA) to incubate at 37 °C for 20 min. Then, the erythrocytes were washed and placed at 37 °C for 1 h after adding 20 μL of Rho(D)Immune Globulin (Human) HyperRHO S/D full Dose (HK-54494, Grifols Therapeutics Inc., Los Angeles, CA, USA). Afterward, the erythrocytes were fully washed with PBS 5 times, and then the cells were resuspended with 1 mL of PBS to prepare 5% of CFSE labeled sensitized erythrocytes suspension (cell density was about 5 × 10^8^ cells/mL). DSP IVIG (final concentration of 62.5 μg/mL) and Fc (final concentration of 5 μg/mL) were added to THP-1 or M1/M2 macrophages cultured in 12-well culture plates. After incubating cells for 24 h, 20 μL of 5% of CFSE labeled sensitized erythrocytes or CFSE labeled non-sensitized erythrocytes were added to each well following co-culturing for 5 h. For adherent cells, 350 μL of trypsin (BD pharmingen, Franklin Lakes, NJ, USA) was used to digest cells, and then PBS containing 2% FBS was appended to terminate the digestion. Afterward, we used 1× lysis buffer (BD pharmingen, Franklin Lakes, NJ, USA) to remove non-phagocytized erythrocytes. After washing cells and discarding the supernatant liquid, retained cells were resuspended with 300 μL of PBS containing 2% FBS for subsequent flow cytometry detection. Lastly, FlowJo V10 software was used to calculate and analyze the phagocytosis rate (phagocytosis rate = phagocytosis rate of macrophages phagocytizing CFSE labeled sensitized erythrocytes − phagocytosis rate of macrophages phagocytizing CFSE labeled non-sensitized erythrocytes).

### 4.6. Cell Apoptosis

Cell apoptosis was detected using the cell apoptosis kit (FA101, Transgen Company, Beijing, China). Thus, 1 mL of THP-1 cells was seeded in 96-well plates (5000 cells per well) and induced into macrophages as described above. Three replicates were set per protein sample (PBS, albumin, Fc, F-IVIG, M-IVIG, Mix-IVIG). After differentiation, the cells were collected and resuspended with 80 μL of 1× Annexin V binding buffer after appending 5 μL of Annexin V and 5 μL of PI to each sample. After reaction at room temperature (20–25 °C) in the dark for 20 min, 300 μL of 1× Annexin V binding buffer was added following flow cytometry detection.

### 4.7. M1/M2 Macrophage Markers

After collecting cells, 20 μL of FITC Mouse Anti-Human CD14 (BD Pharmingen, Franklin Lakes, NJ, USA), 20 μL of PE Mouse Anti-Human CD36 (BD Pharmingen, Franklin Lakes, NJ, USA), 5 μL of PE-Cyanine5 Anti-Hu CD86 (Invitrogen, Waltham, MA, USA), 5 μL of PE-Cyanine7 Anti-Hu CD68 (Invitrogen, Waltham, MA, USA), and 20 μL of APC Mouse Anti-Human CD206 (BD Pharmingen, Franklin Lakes, NJ, USA) were appended to acquiring differentiation status. Significantly, CD206 and CD68 required cell fixation and membrane rupture before staining.

### 4.8. Cells’ FcγRs Expression

Antibodies including 20 μL of FITC Mouse Anti-Human CD16 (BD Pharmingen, Franklin Lakes, NJ, USA), 20 μL of FITC Mouse Anti-Human CD32 (BD Pharmingen, Franklin Lakes, NJ, USA), and 20 μL of FITC Mouse Anti-Human CD64 (555527, BD Pharmingen) were used to satin collected cells. After incubating in the dark at 4 °C for 30 min, the cells were washed and resuspended for flow cytometry detection.

### 4.9. The Ability of Fc Segment of DSP-IVIG to Active Complement

An amount of 125 μL of hematocrit erythrocytes was sucked and washed with PBS 5 times. The supernatants were discarded for the last centrifugation, and the cells were resuspended with 6.25 mL of PBS. Tannic acid (Sigma, St. Louis, MO, USA) was appended to the suspensions following incubation at 37 °C for 15 min. The erythrocytes were washed with PBS and resuspended with 12.5 mL of PBS, and then 1 mL of diphtheria toxoid (Chengdu Rongsheng Pharmaceuticals Co., LTD, Chengdu, China) was acceded to the suspensions following incubation at 37 °C for 30 min. After incubation, the erythrocytes were washed and resuspended with 4.5 mL of bovine albumin barbital buffer. In this study, we adjusted the pH of the IVIG sample to 6.8~7.0. IVIG samples and Human Immunoglobulin for Fc Function BRP (Council of Europe, EDQM) were diluted with bovine albumin barbital buffer to a concentration of 40 mg/mL. Then, we added 100 μL of erythrocytic suspension and 900 μL of 40 mg/mL of IVIG samples together and incubated at 37 °C for 30 min after gently and fully mixing. Subsequently, the cells were washed twice and centrifuged to discard the supernatant and then resuspending cells with 800 μL bovine albumin barbital buffer. Then, 160 μL of erythrocytic suspensions and 40 μL guinea pig complement (Beijing Bai Ao Lai Bo Company, Beijing, China) were appended to every hole of the 96-well plate. The absorbance kinetics were detected at 541 nm within 30 min. Finally, we drew the hemolytic kinetics curves and calculated the activity of Fc segments activating complement (IFc) of IVIG samples with the following formula. In the formula, Sexp was the maximum slope between the adjacent three points of the hemolytic reaction kinetic curve, and As was the original absorbance of each hole.
(1)$$I_{\mathrm{Fc}}=\frac{\frac{Sexp}{As}\;of\;IVIG\;sample-\frac{Sexp}{As}\;of\;Ngative\;control}{\frac{Sexp}{As}\;of\;European\;reference-\frac{Sexp}{As}\;of\;Ngative\;control}\;\times \;\mathrm{biological}\;\mathrm{activity}\;\mathrm{of}\;\mathrm{European}\;\mathrm{standard}\;F_{C}\;\times \;100\%.$$

### 4.10. The Capacity of Fc Segment of DSP-IVIG to Bind FcγRs

First, we collected THP-1 cells and then incubated them in 5 mL of 10% FBS-PBS solution at room temperature for 20 min to block nonspecial binding. After incubation, the cells were centrifugated and resuspended with 0.2% HSA-PBS. An amount of 50 μL of cell suspension and 50 μL of IVIG diluent samples were blended and placed on ice for 1 h. We set three replicates for each IVIG sample. Subsequently, we washed the cells with PBS and then appended 5 μL of FITC labeled Goat Anti-Human IgG Fab fragment antibody (Sigma, St. Louis, MO, USA) following 30 min of incubation in the dark. In the end, after washing the cells 3 times, 5 μL of 7-AAD (BD Pharmingen, Franklin Lakes, NJ, USA) was added to the suspension following incubating for 30 min in the dark for flow cytometry detection. FlowJo 10.0 software was used for data processing.

### 4.11. Statistical Analysis

Data were analyzed by one-way ANOVA followed by Tukey’s multiple comparison tests using GraphPad Prism 7.01 (GraphPad, La Jolla, CA, USA). However, Figure 5B used a 2-way ANOVA followed by a Bonferroni posttest. In all cases, error bars were the standard error of the mean (SEM), and *p* < 0.05 was considered statistically significant.

## 5. Conclusions

In summary, three types of DSP-IVIG can effectively improve the ITP mouse model and cellular phagocytosis model, but the anti-inflammatory effect of M-IVIG was superior to that of F-IVIG and Mix-IVIG. The treatment of severe ITP with IVIG required high-dose usage (1–2 g/kg). If IVIG from a certain sexual plasma source had better therapeutic effects on ITP, or if different DSP-IVIG had disparate advantages in the treatment of various diseases, the clinical dosage of IVIG can be optimized, and then the supply and demand of blood products can be alleviated. Additionally, the phenomenon of IVIG intolerance can be reduced, which was conducive to the establishment of a personalized treatment system for IVIG.

## Figures and Tables

**Figure 1 ijms-24-15993-f001:**
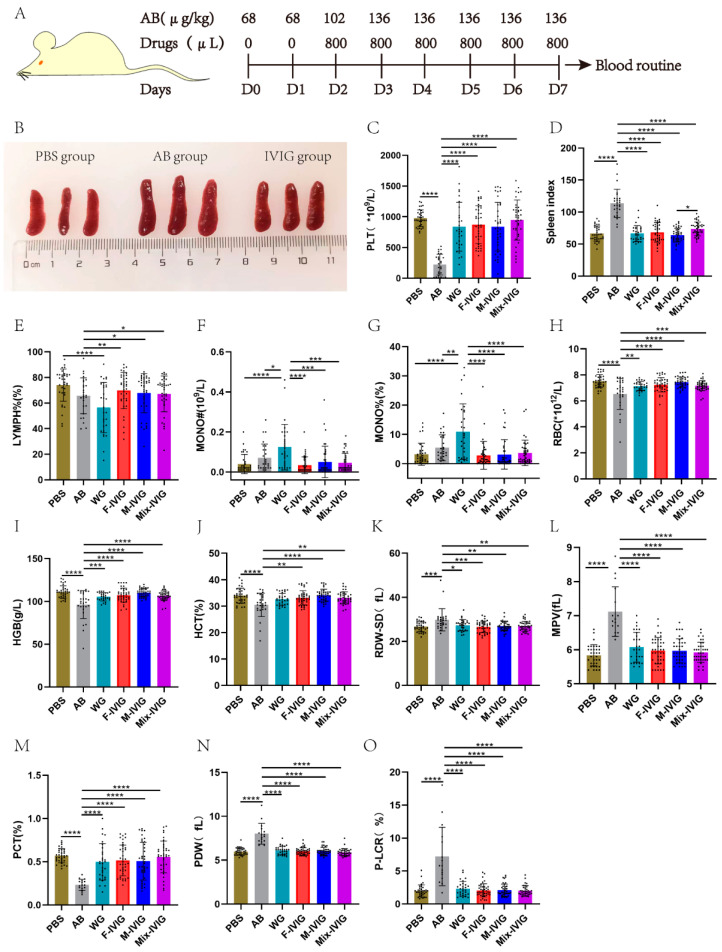
Mouse blood routine test and experiment protocol: (**A**) Mouse modeling protocol. Mice were subjected to the following dosages of MWReg30: 200 μL of 68 μg/kg on D0–D1, 102 μg/kg on D2, and 136 μg/kg on D3–D7. Starting from D3, 2 h after MWReg30 administration, mice received 800 μL of IVIG (50 mg/mL) for treatment. On D7, mice were euthanized, and routine blood tests were conducted. (**B**) Spleen comparison. Comparison of spleen sizes among PBS, AB, and IVIG group mice. (**C**–**O**) Mouse blood routine. PBS represents the mice injected with PBS and albumin. AB was the mice given MWReg30 and albumin. WG, F-IVIG, M-IVIG, and Mix-IVIG represent the mice injected with MWReg30 and WG IVIG, F-IVIG, M-IVIG, and Mix-IVIG, respectively. PLT (platelet), LYMPH% (Lymphocyte (%)), MONO# (Monocyte (#)), MONO% (Monocyte (%)), RBC (Red Blood Cell), HGB (Hemoglobin), HCT (Hematocrit), RDW-SD (Red Blood Cell Distribution Width—SD), MPV (Mean Platelet Volume), PCT (Plateletcrit), PDW (Platelet Distribution Width), P-LCR (Platelet—Large Cell Ratio). Sample sizes (N) were greater than 10. * *p* < 0.05, ** *p* < 0.01, *** *p* < 0.001, and **** *p* < 0.0001.

**Figure 2 ijms-24-15993-f002:**
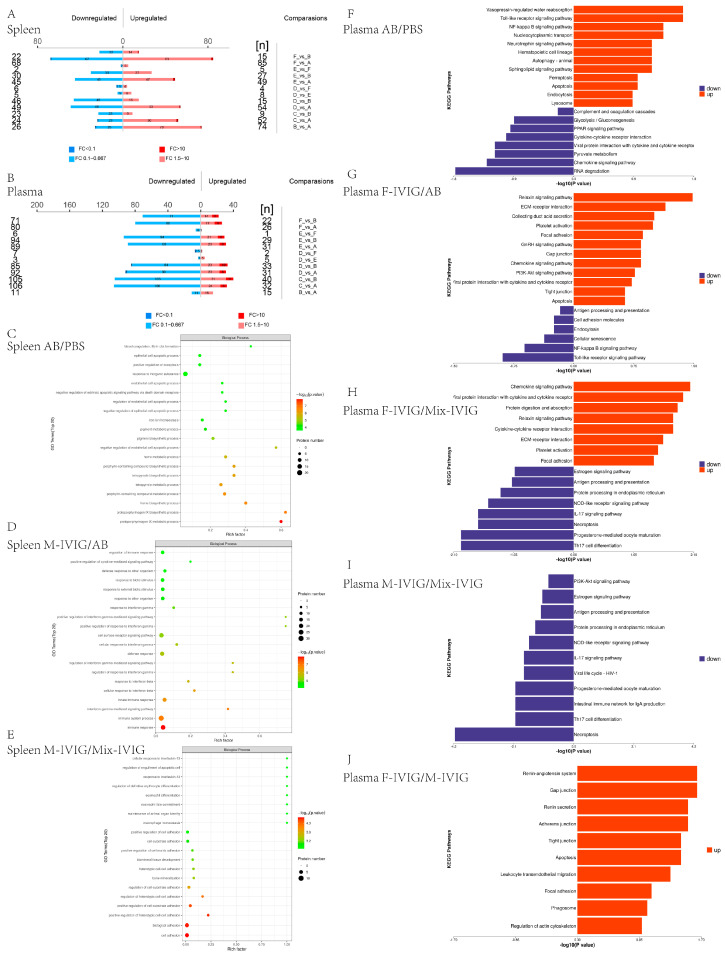
Mouse Proteomics: (**A**,**B**) Numbers of differentially expressed proteins in mouse spleen and mouse plasma. Comparisons are different comparison groups; upregulated is upregulated proteins, while downregulated means downregulated proteins. (**C**–**E**) GO analysis. The abscissa is the rich factor (the proportion of the number of differentially expressed proteins annotated to a GO functional category to the total number of identified proteins annotated to that GO functional category). The vertical axis represented the statistical results of differential proteins under each GO classification. Bubble color means significance of enriched GO classification (*p*-value calculated based on Fisher’s Exact Test). Color gradient is the size of the *p*-value (−log10). The closer the color is to red, the smaller the *p*-value and the higher the significance level of enrichment under the corresponding GO classification. (**F**–**J**) KEGG pathway. The abscissa represents the *p*-value, while the vertical axis reveals the pathways’ names. Red (right) represents the upregulation signaling pathway, and blue (left) is the downregulation signaling pathway. PBS represents the mice injected with PBS and albumin. AB is the mice given MWReg30 and albumin. WG, F-IVIG, M-IVIG, and Mix-IVIG represent the mice injected with MWReg30 and WG IVIG, F-IVIG, M-IVIG, and Mix-IVIG, respectively.

**Figure 3 ijms-24-15993-f003:**
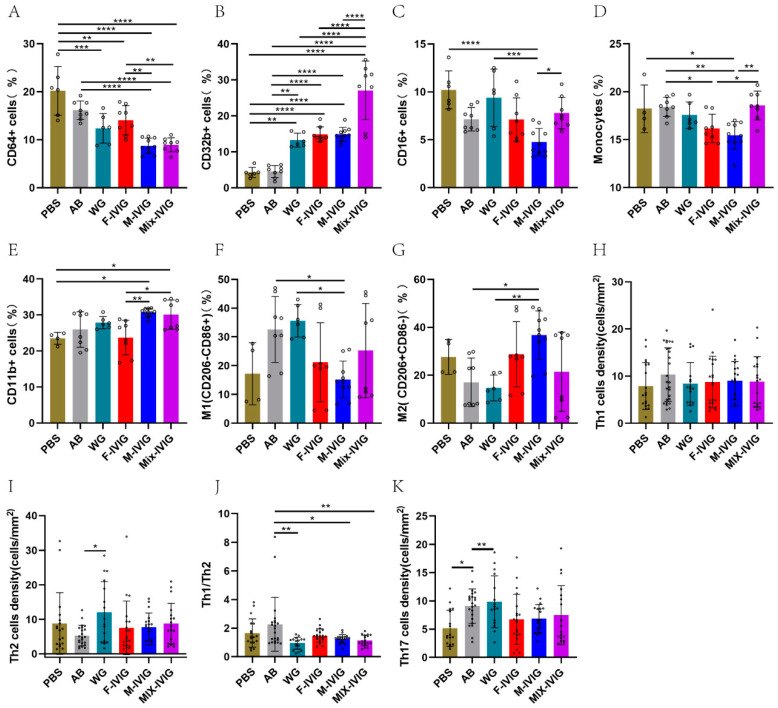
Mouse spleen cell expression: (**A**–**C**) Mouse FcγRs expression status. (**D**–**G**) M1 and M2 macrophages in mouse spleen. (**H**–**K**) The expression of Th1, Th2, and Th17 in mouse spleen. PBS represents the mice injected with PBS and albumin. AB is the mice given MWReg30 and albumin. WG, F-IVIG, M-IVIG, and Mix-IVIG represent the mice injected with MWReg30 and WG IVIG, F-IVIG, M-IVIG, and Mix-IVIG, respectively. Sample sizes (N) greater than 3. * *p* < 0.05, ** *p* < 0.01, *** *p* < 0.001, and **** *p* < 0.0001.

**Figure 4 ijms-24-15993-f004:**
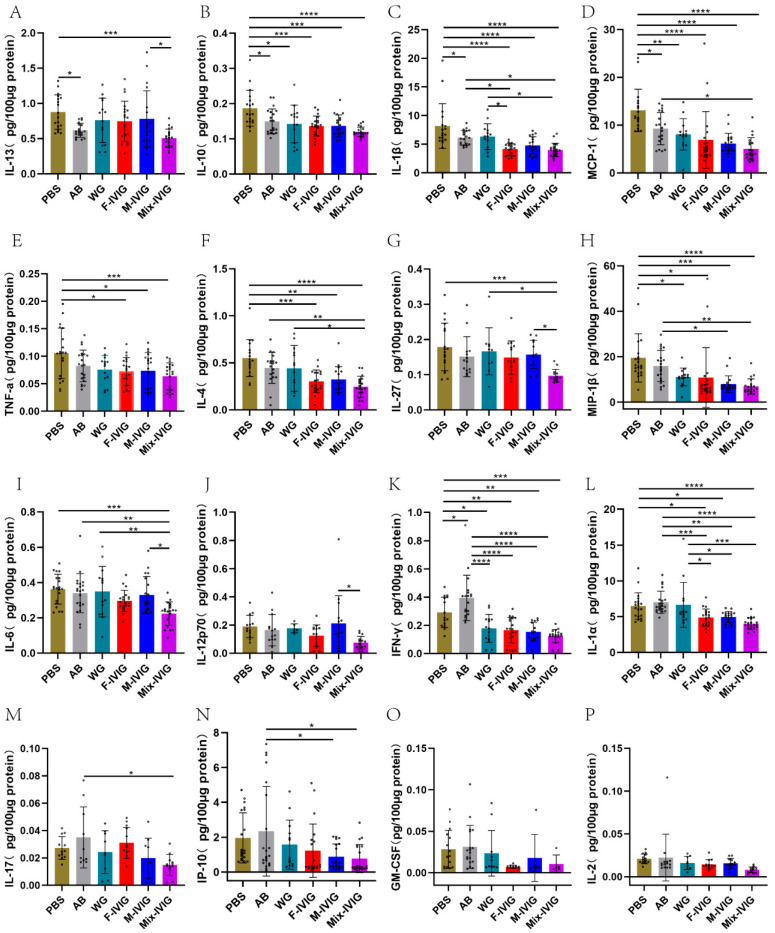
Expression of cytokines in mouse spleen: (**A**–**P**) The ordinate represented IL-3 (Interleukin-3), IL-10 (Interleukin-10), IL-1β (Interleukin-1β), MCP-1 (Monocyte Chemoattractant Protein 1), TNF-α (Tumour Necrosis Factor-α), IL-4 (Interleukin-4), IL-27 (Interleukin-27), MIP-1β (Macrophage Inflammatory Protein 1β), IL-6 (Interleukin-6), IL-12p70 (Interleukin-12p70), IFN-γ (Interferon-γ), IL-1α (Interleukin-1α), IL-17 (Interleukin-17), IP-10 (Interferon γ-induced protein), GM-CSF (granulocyte–macrophage colony-stimulating factor), and IL-10 (Interleukin-10), respectively. PBS represents the mice injected with PBS and albumin. AB is the mice given MWReg30 and albumin. WG, F-IVIG, M-IVIG, and Mix-IVIG represent the mice injected with MWReg30 and WG IVIG, F-IVIG, M-IVIG, and Mix-IVIG, respectively. Sample sizes (N) are greater than 10. * *p* < 0.05, ** *p* < 0.01, *** *p* < 0.001, and **** *p* < 0.0001.

**Figure 5 ijms-24-15993-f005:**
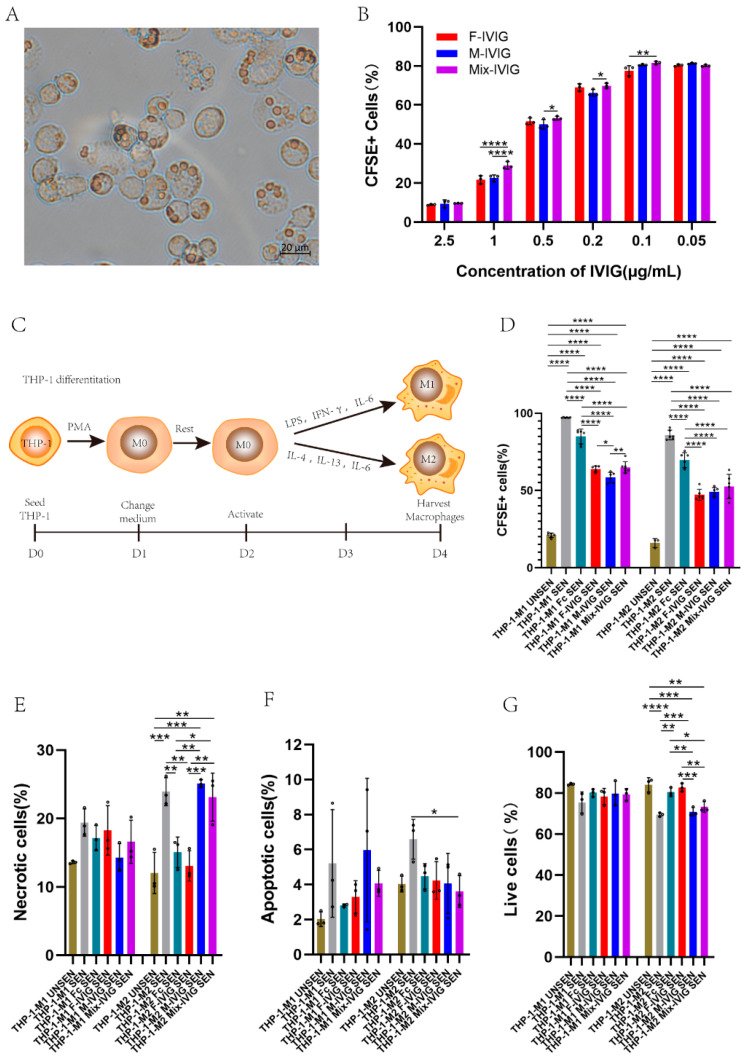
The effects of DSP-IVIG on macrophages: (**A**) Microscopic observation of macrophage phagocytosis of sensitized erythrocytes. (**B**) Inhibition of THP-1 phagocytosis by DSP-IVIG. (**C**) Methods for inducing THP-1 into M1 and M2 macrophages. PMA added on day 1 (D1), induction into M1 macrophages with LPS, IFN-γ, and IL-6 on D2, and induction into M2 macrophages with IL-4, IL-13, and IL-6 on D2. Cells collected on D4. (**D**) Inhibition of THP-1 M1/M2 phagocytosis by DSP-IVIG. (**E**–**G**) Cell apoptosis status. THP-1-M1/M2 UNSEN means M1/M2 derived from THP-1 following the addition of insensitive erythrocytes, while THP-1-M1/M2 SEN is the addition of sensitive erythrocytes. THP-1-M1/M2 Fc SEN, Female SEN, Male SEN, Mix SEN is M1/M2 derived from THP-1 with added Fc, F-IVIG, M-IVIG, Mix-IVIG, respectively, following adding sensitive erythrocytes. N = 3, * *p* < 0.05, ** *p* < 0.01, *** *p* < 0.001, and **** *p* < 0.0001.

**Figure 6 ijms-24-15993-f006:**
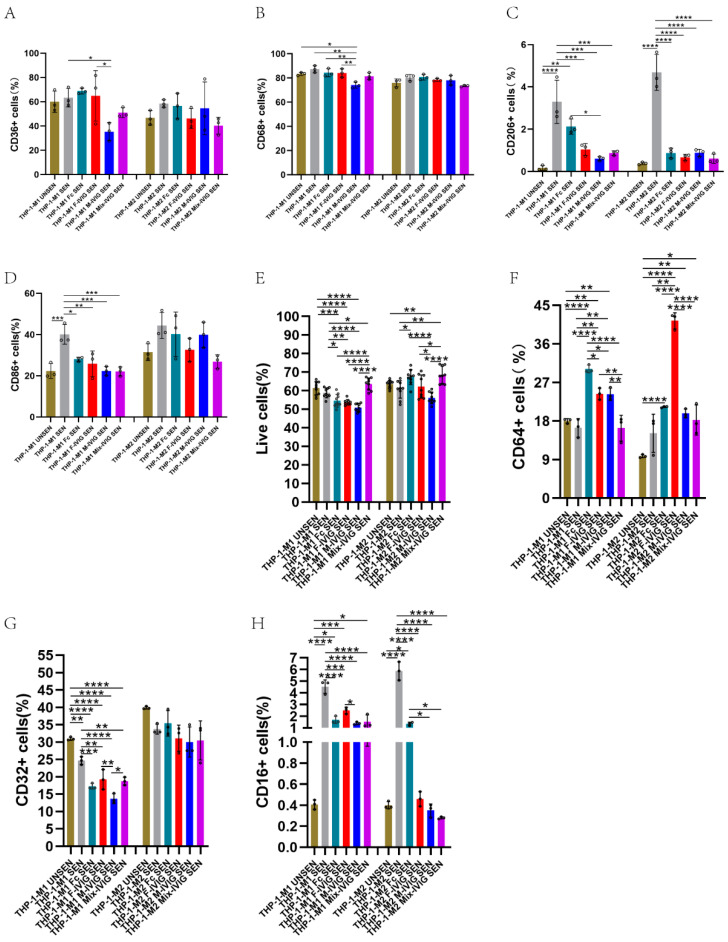
The effects of DSP-IVIG on M1/M2 macrophages and FcγRs: (**A**–**D**) The expression of CD36, CD68, CD86, and CD206 of macrophages. (**E**–**H**) The expression of FcγRs. (**E**) The number of living cells. (**F**–**H**) The expression of CD64, CD32, and CD16 of macrophages. THP-1-M1/M2 UNSEN is M1/M2 derived from THP-1 following the addition of insensitive erythrocytes, and THP-1-M1/M2 SEN is the addition of sensitive erythrocytes. THP-1-M1/M2 Fc SEN, Female SEN, Male SEN, Mix SEN was M1/M2 derived from THP-1 with added Fc, F-IVIG, M-IVIG, Mix-IVIG, respectively, following the addition of sensitive erythrocytes. N = 3, * *p* < 0.05, ** *p* < 0.01, *** *p* < 0.001, and **** *p* < 0.0001.

**Figure 7 ijms-24-15993-f007:**
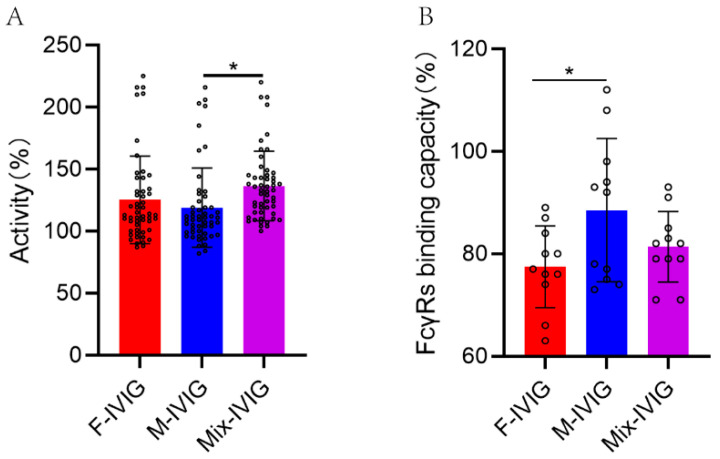
The ability of the Fc segment of DSP-IVIG to activate complement and to bind FcγRs: (**A**) The ability of Fc segment of DSP-IVIG to activate complement. (**B**) The capacity of Fc segment of DSP-IVIG to bind FcγRs. N > 10, * *p* < 0.05.

**Table 1 ijms-24-15993-t001:** Groups of ITP mice. PBS means the mice were injected with PBS and albumin. AB was the mice given MWReg30 and albumin. WG, F-IVIG, M-IVIG, and Mix-IVIG represented the mice injected MWReg30 and WG IVIG, F-IVIG, M-IVIG, and Mix-IVIG, respectively.

Groups	ITP Model	Treatment
PBS	PBS	Albumin
AB	MWReg30	Albumin
WG	MWReg30	WG IVIG
F-IVIG	MWReg30	F-IVIG
M-IVIG	MWReg30	M-IVIG
Mix-IVIG	MWReg30	Mix-IVIG

## Data Availability

The mass spectrometry proteomics data have been deposited to the ProteomeXchange Consortium (http://proteomecentral.proteomexchange.org, accessed on 26 September 2023) via the iProX partner repository with the dataset identifier PXD045681.

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
