# Peer review of "Study on the Treatment of ITP Mice with IVIG Sourced from Distinct Sex-Special Plasma (DSP-IVIG)"

_ijms, 2023, doi:10.3390/ijms242115993_

Round 1
Reviewer 1 Report
Comments and Suggestions for Authors
The article entitled “Study on the treatment of ITP mice with IVIG sourced from distinct sex-special plasma (DSP-IVIG)” addresses an original topic regarding the efficacy of IVIG plasma sourced from male plasma donor , female plasma donor, or male/female plasma donor. The project work is well conducted and the informations about the immunology of ITP are well described and show that the authors have an expertise in this area. Infact, the authors use a mice model of ITP and investigate a great panel of inflammatory markers (immune cells, cytokines, FcγRs) in association with a monocyte-macrophage inflammation model in order to investigate the presence of pro-inflammatory M1 cells or anti-inflammatory M2 cells. The results of this study show that the IVIG derived by male plasma has more anti-inflammtory efficacy than IVIG derived by female or male/female. In addition, the IVIG derived from male plasma donors improve more effectively the spleen index than the IVIG obtained from female or male/female plasma donors. I think that the informations of this study are important in all setting characterized by autoimmune diseases such as hematology, rheumatology, neurology, internal medicine. Therefore, I think that this article is suitable for publication in its current version.
Author Response
Based on your opinion, there is no need to make revisions to the manuscript. Thank you for reading the manuscript and your valuable comments.
Reviewer 2 Report
Comments and Suggestions for Authors
Authors examined the ability of DSP-IVIG to activate complement and the binding ability to FCγRs on the surface of THP-1
Although this manuscript is potentially interesting, several issues arise.
1) Abstract is too long. Abstract should be summarized.
2) Although experiments are interesting, it is hard to understand these experiments.
3) Figures and Tables have many abbreviations. These abbreviations should be explained.
4) Additional experiment with male mice may be helpful.
5) This experiment seems to be no significant difference of platelet count between F-IVIG and M-IVIG.
6) If the object will be treatment ITP mice with DSP-IVIG, improvement of platelet count should be shown by the treatment with DSP-IVIG.
7) Figures are too busy and not clear. Figures should be remade.
Round 2
Reviewer 2 Report
Comments and Suggestions for Authors
Authors fully responded to my comments.
I have no further comments.